# Understanding Trustful Relationships between Community Health Workers and Vulnerable Citizens during the COVID-19 Pandemic: A Realist Evaluation

**DOI:** 10.3390/ijerph19052496

**Published:** 2022-02-22

**Authors:** Dorien Vanden Bossche, Sara Willems, Peter Decat

**Affiliations:** Department of Public Health and Primary Care, Faculty of Medicine and Health Sciences, Ghent University, 9000 Ghent, Belgium; sara.willems@ugent.be (S.W.); peter.decat@ugent.be (P.D.)

**Keywords:** community health workers, primary healthcare, vulnerable populations, trust, COVID-19, realist evaluation, grounded theory

## Abstract

(1) Background: Community health workers (CHWs) are an essential public health workforce defined by their trustful relationships with vulnerable citizens. However, how trustful relationships are built remains unclear. This study aimed to understand how and under which circumstances CHWs are likely to build trust with their vulnerable clients during the COVID-19 pandemic. (2) Methods: We developed a program theory using a realist research design. Data were collected through focus groups and in-depth interviews with CHWs and their clients. Using a grounded theory approach, we aimed to unravel mechanisms and contextual factors that determine the trust in a CHW program offering psychosocial support to vulnerable citizens during the COVID-19 pandemic. (3) Results: The trustful relationship between CHWs and their clients is rooted in three mental models: recognition, equality, and reciprocity. Five contextual factors (adopting a client-centered attitude, coordination, temporariness, and link with primary care practice (PCP)) enable the program mechanisms to work. (4) Conclusions: CHWs are a crucial public health outreach strategy for PCP and complement and enhance trust-building by primary care professionals. In the process of building trustful relationships between CHWs and clients, different mechanisms and contextual factors play a role in the trustful relationship between primary care professionals and patients. Future research should assess whether these findings also apply to a non-covid context, to the involvement of CHWs in other facets of primary healthcare (e.g., prevention campaigns, etc.), and to a low- and middle-income country (LMIC) setting. Furthermore, implementation research should elaborate on the integration of CHWs in PCP to support CHWs in developing the mental models leading to build trust with vulnerable citizens and to establish the required conditions.

## 1. Introduction

Community health workers (CHWs) are an essential public health workforce defined by their trustful relationships with communities facing economic and social disadvantages [1]. They are laypeople who share lived experiences and have social capital in their communities. Although their job titles vary worldwide, there is agreement that, through a brief training, their core roles include health outreach, education, cultural mediation, and advocacy, among others [2]. As well as for chronic conditions [3,4,5,6], as in acute situations, like infectious disease outbreaks, the use of CHWs has shown to be effective [7,8]. During the recent COVID-19 pandemic, CHWs have demonstrated their valuable role in supporting public health efforts [9,10,11] by providing health education and outreach to community members and by promoting culturally appropriate preventive behaviors, contact tracing, and data collection, limiting the disease’s spread, and in addressing mental health issues [12,13,14,15,16].

Considerable evidence confirms that CHWs’ role in enabling vulnerable communities to access care is underpinned by the trustful relationships with their clients [17,18]. In addition, existing literature states that CHWs perceive ‘building trust with the community’ (beyond trust in health services) as a critical component of their practice [19,20,21,22,23,24]. However, it remains unclear how these trustful relationships are built, or in other words, which mechanisms lead to the building of this trust.

Trust, a complex, multifaceted notion, ‘influences individuals’ willingness to act based on others’ words, motives, intentions, actions and decisions under conditions of uncertainty, risk or vulnerability’ [25]. Social relations of trust are accepted as a core contributor to health systems’ performance; a trust-based health system is grounded in cooperation, communication, and empathy, enabling the successful functioning of the health service [26]. Trust is a complex construct, of which the specific elements vary between settings and relationships. The conceptual framework by Gilson et al. [27] describes the interaction between workplace and patient–provider trust (see Figure 1). Workplace trust, defined as respectful and fair treatment in the workplace, is rooted in trust in the employing organization, the supervisor, and colleagues. Patient–provider trust is rooted first in interpersonal trust, in this case between a patient and CHW. Patient–provider trust is also rooted in institutional trust, the extent to which the CHW and patient can trust that the health system will support the CHWs to act in the patient’s best interests. Here, the health system refers in an immediate sense to the CHW’s coordinators, the primary care professionals at the primary care practice (PCP), as well as the broader health system [18]. For health systems, therefore, the elements of workplace trust may provide the basis for identifying the managerial action required to improve patient–provider trust and, therefore, responsiveness. The conceptual framework suggests that healthcare providers’ trust in their workplace influences their attitudes and behaviors towards patients in ways that then influence the extent of trust between patients and providers. Trust is, therefore, both an input/independent variable (in the form of workplace trust) and an output/dependent variable (in the form of patient–provider trust) in this framework, and health system performance is critically dependent on these relationships. The framework also acknowledges that patient–provider trust is influenced by the behaviors of both parties to the relationship [27]. Finally, the figure suggests that the interaction between workplace trust and patient–provider trust may be influenced by three other factors: (1) workers’ personalities and past experiences; (2) the broader interactions between providers and the community they serve; and (3) features of the historical, cultural and socio-political context of the health system [27].

The present study aimed to provide insight into the meaning of trustful relationships between CHWs and vulnerable citizens and to understand how and under which circumstances these CHWs are likely to gain trust in their communities. A realist method of evaluation was undertaken in order to answer the following questions: (1) Which mechanisms enable CHWs to gain trust in a vulnerable patient community and to establish psychosocial health outcomes, and which context factors influence this trust-building between CHWs and vulnerable citizens? (2) What makes the CHW approach complementary to the interaction with formal caregivers in primary healthcare settings? The conceptual framework by Gilson et al. [27] is used as a theoretical framework to guide the analysis. Our purpose is to understand how interpersonal trust between CHWs and vulnerable citizens plays out in the workplace of PCP in Ghent during the COVID-19 pandemic, and from this make recommendations on the building of trustful relationships with community members in other high-income country (HIC) primary care settings.

## 2. Materials and Methods

This paper reports on a qualitative evaluation of a support program by CHWs in the city of Ghent. The quantitative evaluation of this program showed that recipients experienced a significant improvement in a self-rated impression of change in psychosocial health and were highly satisfied with the intervention. These results were reported elsewhere [15].

### 2.1. Study Setting

The setting of this study was the city of Ghent, the third-largest city in Flanders, Belgium (266,000 inhabitants). Since April 2020, a team of 30 CHWs has offered psychosocial support at home to vulnerable people at risk of becoming victims of fear and social isolation during the COVID-19 pandemic. This program was set up in collaboration with the Department of Welfare and Health of the city of Ghent and was run and coordinated by L.V., who works as a social worker at this same department.

All CHWs were trained using 2 online training modules of 2 h, entailing communication skills, providing correct information, recognizing alarming signals presented by clients, and safety measures to prevent COVID-19 infection. Additional on-demand support was provided, and peer-to-peer coaching was provided in small groups once a month.

Primary care professionals identified eligible patients for whom they saw a role for the CHW. Eligible patients (1) had a limited social network; (2) were older than 18 years; and (3) had a background of vulnerability. Vulnerability was defined in this program by the research team as having a psychiatric history, or a precarious social context, or an uncertain residence status, or a chronic illness, or going through a recent critical event such as bereavement or divorce, or being older than 65 years. The first contact between CHW and the client was always made by phone. In this first contact, CHWs presented themselves, checked in with how their clients were doing, and explored how their assigned clients wanted to organize the following contacts. Further on, the CHWs communicated with clients regularly, depending on clients’ expressed needs.

CHWs provided hands-on, tailored support to clients spanning the domains of social support, coaching, advocacy, and navigation to healthcare if needed. The overall goal was to offer presence to socially isolated clients or those who felt lonely or anxious. By being present, CHWs offered a sympathetic ear and gave attention to their clients’ worries, stories, and questions. CHWs were also instructed to check whether their clients were correctly informed about the most recent distancing measures. If this was not the case, the CHWs provided and explained the updated preventive measures. Moreover, when clients presented with alarming signals according to their psychosocial state, CHWs took responsibility to inform their caretakers and the coordinating team about the situation. Contacts were via text messages, WhatsApp messages, emails, postcards, Skype meetings, phone calls, or face-to-face meetings. Contacts took between a few minutes several hours. CHWs did not directly provide health education or clinical care, and when these needs arose, CHWs navigated clients to the appropriate healthcare provider. Intervention guidelines are codified in detailed manuals, in-person and online training, documentation, and reporting.

### 2.2. Ethics

Ethical approval for this study was obtained from the Ethics Committee (EC) of the Ghent University Hospital (EC registration number: BC-07744).

### 2.3. Study Design

In this qualitative study, we adopted a realist evaluation method. Realist evaluation studies have the purpose of identifying ‘what works in which circumstances and for whom?’ [28]. Therefore, realist evaluation aims to identify the hidden causal forces behind empirically observable patterns or changes in those patterns [29]. This is done through ‘retroduction’: going back from observed patterns and looking below the surface for what might have produced them. As such, realist studies focus on context and necessary conditions for generating social mechanisms, making it a practical approach for studying complex social issues such as psychosocial health and building trust [30]. The central research question of this project is realist in nature: it aims to identify underlying mechanisms and the circumstances in which these mechanisms emerge. The theory developed as an answer to this research question is subject to further refinement and enrichment in consequent research loops, which is characteristic of realist evaluation. In this study, the realist evaluation formed the logic of inquiry and theory was developed following the principles of grounded theory [28,31]. We chose this approach because of the appreciation of realism as a ‘logic of inquiry that generates distinctive research strategies and designs, and then utilizes available research methods and techniques within these’ [28]. Realist evaluation starts from an Initial Programme Theory (IPT) formulated in a context-mechanism outcome-structure (CMO). In this study, we started from an IPT, tracing processes backward to study the question ‘what is it about CHWs that allows them to build trust with vulnerable communities, why is that, and under which circumstances?’. The IPT in this study was drawn from a literature review and from input we collected from the CHWs and the project coordinators in the course of the intervention. In addition, the Gilson framework was used as a structure to build the IPT and to guide the analytical steps. The IPT was formulated as follows: ‘CHWs can build trust with vulnerable citizens during the lockdown and physical distancing measures in a primary care setting through the following mechanisms: by offering a sympathetic ear and standing close to the community, CHWs can profile as accessible persons, allowing vulnerable clients to feel connected with others and to feel less lonely.’. The IPT was then tested in focus group discussions with CHWs and in in-depth interviews with CHWs and clients. Testing the hypotheses in the IPT, sourcing from the already existing Gilson framework, led to the development of a middle-range theory (MRT), consisting of CMO hypotheses on how, why and when CHWs can build trust with vulnerable communities. A MRT is a program theory, developed at a mid-level range of abstraction, i.e., a theory concrete enough to test yet generalizable to different contexts. This program theory clarifies why, how, and in which circumstances CHWs can build trustful relationships with vulnerable clients (referring to mechanisms and influencing context factors of improved psychosocial health outcomes) [32]. This MRT can be tested and refined through further studies.

### 2.4. Recruitment of Participants, Data Collection, and Data Analysis

To collect data for this qualitative study, focus group discussions and in-depth interviews were organized. All participants agreed to participate and gave informed consent.

For the focus group discussions, all 30 CHWs were invited face-to-face and via email by the project coordinators and the researchers. These focus groups aimed to discuss, validate and adapt the IPT presented. In order to accede to different CHW profiles, the focus groups were organized at different times, i.e., one in the afternoon and one in the evening. The focus group discussions were semi-structured, using an interview guide based on the IPT. The researchers of the research team discussed the guide. Afterward, the research team identified and explored gaps, contradictions, and uncertainties in the data.

Then, in-depth interviews were conducted with a purposive sample of CHWs and clients. When selecting CHWs and clients for these in-depth interviews, we aimed for respondents to represent a maximal variation in age, gender, ethnic background, and occupation. All in-depth interviews were semi-structured, using an interview guide based on the IPT and on the data of the focus group discussions (complete interview guides available in Appendix A). The researchers of the research team discussed the guide. They revised it after two test interviews. The interviewer (D.V.B.) started by asking about any psychosocial health-related effects participants experienced and then asked how participants think these effects come about and which context factors are necessary for allowing these effects to occur. Where possible, interesting data from previous interviews were discussed and refined in later interviews. Once theoretical data saturation [33] was reached for in-depth analysis, we stopped including new participants. Field notes were taken during the interviews.

The focus group discussions and in-depth interviews were audio-recorded, pseudonymized, and transcribed verbatim. The data were structured following the CMO categories of the realist evaluation approach, using a grounded analysis [34]. Four analytical steps were taken: (1) open coding of data was used to identify new concepts and to discover their properties and dimensions (meaning without using a predefined coding tree); (2) axial coding was used to construct linkages between the data, categories and subcategories were formed; (3) selective coding of data explicated the interconnection of these categories and subcategories [31]; and (4) these interconnecting categories and subcategories were structured in a program theory. This was extensively discussed with the supervisors (P.D. and S.W.) and finally led to an adapted concluding theory [35].

## 3. Results

### 3.1. Study Participants

First, two focus groups (*N*  =  4 and *N*  =  4) were organized (September 2020) with CHWs. Then, in-depth interviews were conducted with a purposive sample of 13 CHWs and 11 clients (September–November 2020). Sociodemographic descriptors for the participating CHWs and clients are presented in Table 1. Some of the CHWs who participated in the focus group discussions were interviewed individually afterward.

In the following sections, the CMO configurations that resulted from the third analytical step are described. Finally, the discussion explicates how the CMO configurations are linked together in an overall program theory.

### 3.2. Outcomes of the Intervention

Participants agreed on the fact that the intervention was needed at the time it was introduced. The participants expressed positive feelings towards the intervention and its **influence on feelings of self-worth and a sense of belonging**. Clients felt like they were worth the effort of being listened to, “e.g., I get a call, so I exist”, and felt a social connection with others in times of physical distancing and quarantining, which made them feel less lonely.


*She also said, ah, I am not alone. Someone gave my phone number to someone. I am not alone in life, after all. We are human, and we need this.*
(CHW 1)


*Then I thought, “Yes, I see someone.” I had not seen anyone for three weeks, and suddenly someone was standing there smiling for you, and he just asked: “Can I do something for you?” ... That gives the feeling of “Yes, I count in this society!”. And look, just, someone standing at your door, a laugh.*
(Client 1)

Participants reported that the **CHW activated them to find a moment of relaxation and distraction**. For a while, they could forget about the continuous threat of COVID-19.


*I felt that it was a relief for her that there was someone just to eat together with or take a walk.*
(CHW 2)


*The first time I remember well, she had a bowl of strawberries with her. And we ate half of it. And we have been chatting a bit about my kids. Time flew by.*
(Client 3)

Participants mentioned that they **felt renewed energy by having a positive new experience**. They felt that they could charge their batteries and build some reserve to face further the challenges they were confronted with. Also, they were touched by this small gesture, which gave them a “zest for life”.


*We talked about everything. We went for a walk 3 or 4 times. I was happy, and I needed that. I had to walk and move. I needed air. I needed a slight push, and she was. After that, I also started exercising several times.*
(Client 2)

### 3.3. The Program Mechanisms That Produce Outcomes

The above outcomes of the CHW intervention have a positive impact on the psychosocial health of vulnerable clients. A central driver for these outcomes is the trust built by CHWs in their interaction with vulnerable clients. The identified program mechanisms driving the building of trustful relationships between CHWs and clients can be grouped into three distinct but overlapping themes: recognition, equality, and reciprocity. The interaction of program mechanisms and contextual factors to build trust are illustrated in Figure 2.

#### 3.3.1. Recognition

**CHWs thoroughly recognized the vulnerability of clients.** Being experienced experts in certain aspects of vulnerability enabled the CHWs to recognize the problematic situations of their clients with a vulnerability background. This experience with vulnerability made CHWs feel capable of handling the responsibility of guiding these clients facing multiple challenges during the lockdown and physical distancing measures. However, some CHWs did not have this vulnerable background, and some of them also mentioned feeling less capable and less confident to carry out this task.


*I also have a background. I am actually an expert in poverty and social exclusion. I have experienced well, by being honest myself and by trying to create trust, that it is a nice interaction. I think it helps me very well. I’ve been through a lot of things myself, and I think people feel that when you’re with them. It’s relatable. They see that, and they feel that.*
(CHW 3)


*I don’t mind that at all, but I really don’t have any experience with that at all. Sometimes I felt a bit incompetent. I did know that a social worker also came to visit him. But I sometimes felt more like I was sitting in the chair as the psychiatrist (nervous chuckle), and that I had to hear him out about his problems, but that I didn’t know what to ask for or what to say.*
(CHW 4)

**Having a broad view of a person’s context makes the picture completer.** By recognizing the various elements that shape an individual’s context, CHWs are able to develop a broad perspective and a realistic overview of one’s situation.


*For example, by going with that person to a hospital. And while waiting, when the door is open, you can already see that this person has probably had a hard time due to Corona. Or when you walk to the bus together with that person, while talking, then you hear that that person didn’t eat that day. That person doesn’t have one problem; health. But that person has so many other problems.*
(CHW 3)

#### 3.3.2. Equality

**CHWs and clients developed a real connection, from human to human.** Almost all participants stated that CHWs were offering “something else” compared to formal caregivers. Many participants identified that this aspect of real human contact is crucial for the CHW intervention. By contrast, patients mainly see professional caregivers as high-ranked experts responsible for their patients’ health and wellbeing.


*I think it’s mostly the fact that ordinary people can help. That she also felt “That’s not a doctor or nurse here but just someone who calls to ask how I’m doing”.*
(CHW 2)


*I’m voluntary. I only come to listen and to chat. That is something different from what you (professional caregivers) can already give to that lady. I’ll give something else. I am a human being, she tells what is on her mind. And I try to listen and say something once at a time—but in the meantime, I’m not deciding about her life.*
(CHW 1)


*I think the big difference lies in the fact that the contact is from person to person. There’s nothing attached to it. It costs zero euros. Nothing is obligatory. It’s not a mandatory therapy and you shouldn’t get anywhere. It is non-committal. It is authentic. I think there is a very big difference there, that you have that buffer. That you are not immediately professionally valued as “Something is wrong with you”.*
(CHW 5)

**CHWs are “on the same level” as the clients they guide**. This aspect of equality enables CHWs to gain trust. Moreover, the difference in social status and ‘not being on the same level’ between formal caregivers and their vulnerable patients can sometimes make them less accessible for people with a vulnerable background to share their full stories for fear of not being understood, as stated by many participants. ‘Not being part of the system’ can also foster trust by a feeling of safety. Being clear to clients about their volunteering status, CHWs can also set proper expectations of their role and unconditional commitment.


*The doctor-patient relationship is something else. They’re higher. She just says X and I say Y. There is no title. I mean, there’s no hierarchical difference. We just sit on a bench together to chat.*
(CHW 6)


*I think that does give more confidence. Because we’re just going somewhere next to and with those people. And sometimes it scares people off when you say “I’m a doctor”. I think people might automatically start to conceal things or their lifestyle. I have found that people speak more easily. Like when walking: we walk next to each other, it is quiet and automatically they tell something more. Then sometimes things come up.*
(CHW 3)

**An unprejudiced stranger can offer a safe interaction to share personal stories and concerns without judging.** Some participants expressed that CHWs are neutral and can offer a new external perspective that can be a breath of fresh air in often complex cases.


*I think the barrier is smaller for some people. That they can tell someone else about their concerns instead of their own children or their partner, who have already heard it all. I think sometimes it’s easier to be open to people who aren’t that close to you.*
(CHW 2)


*Another person (an outsider) just listens and another person will not draw any conclusions or judge.*
(CHW 7)

#### 3.3.3. Reciprocity

**Sharing creates a bond.** CHWs mentioned that they often needed to share their own experiences and stories to create an atmosphere of mutual openness and trust. By contrast, formal caregivers, in general, maintain a professional distance.


*That creates a bond. They say something, and you say something. That makes us equal.*
(CHW 8)

By sharing similar experiences, problems can be recognized, making clients feel less alone in their situation. In addition, clients recognize aspects of the CHWs’ background and vice versa. This familiar sense, which is fundamental to working with volunteers from the community, emerges with a feeling of authenticity.


*Yes, there were similar things. And that she also heard that I was also afraid, I was also scared. No one knew in the beginning what that Corona was. And yes, I know very well what it is to be alone. I’m actually alone too... That one woman I mentored, I really did feel that pain from her. And I think it’s because of that, because I knew what it was, that it was possible to listen to her, to let her tell and to gain confidence. Because I also once dared to say: I know what it is, to be alone. You still feel safer to talk further. I think that’s a surplus.*
(CHW 3)

### 3.4. The Contextual Factors That Support the Program Mechanisms

The contextual factors that were critical in supporting the program mechanisms of this CHW intervention were:Adopting a client-centered attitudeCoordination of the interventionThe temporariness of the interventionLink with the primary care practice

**Adopting a client-centered attitude** was an essential condition to facilitate this process of creating trust. This involves specific CHW attitudes, like demonstrating empathy, compassion, and acting as a soundboard for other people ventilating their emotions while taking the demanded time for this. In addition, an important condition was that CHWs had the ability to balance reciprocity in the contacts while focussing attention on the clients and adapting to the personal needs and boundaries of their clients.

**The coordination of this intervention** by a coordinating team was identified as an essential element for a successful intervention. CHWs were allowed a relative amount of personal freedom in filling in the details of their role. They differed in ideas on responsibilities and boundaries, and, hence, performance and concrete actions. This free interpretation requires a high level of trust and support by the coordinating team and a willingness of all actors to work with an unfixed and dynamic concept. Participating CHWs reported that they needed this support to feel confident and capable of fulfilling their task. Nevertheless, guidance could have been more intensified for some CHWs, as they sometimes felt overwhelmed by and responsible for the complex problems and situations they were facing.


*I think you have stronger foundations. First, you know there are people standing next to you. The case will be resolved. Then, you are already feeling more secure to go to that person next time. The more information you get, the more knowledge and insight you get.*
(CHW 9)

**The temporary nature** of the intervention enabled the CHWs to make their role a feasible commitment. For various reasons, the COVID-19 pandemic allowed people to spend time on volunteering activities, like being a CHW. The role of CHWs to offer psychosocial support to vulnerable people with limited social networks and suffering from loneliness and anxiety was sometimes mentioned to be demanding. The temporary nature of the intervention made this commitment realistic for volunteers, who have other obligations to fulfill in normal circumstances, when not facing a pandemic situation. The short time span of the intervention also supported the CHW to authentically ‘just be himself/herself’ and deliver the demanded time and flexibility. Of course, when CHWs and clients both wanted to stay in touch, this was possible.


*This way, it is possible for me to keep it bearable in terms of time use. If I were there for three to four hours, that would be too much for me. And then maybe, from what I think I can add, I would lose my strength.*
(CHW 8)

**The collaboration with primary care practice** showed to be an essential condition for building trust for both CHWs and clients. Because primary care professionals selected patients who could benefit from the intervention, CHWs had the impression that the clients they were guiding were all open to receiving the intervention. Moreover, clients who had received information concerning the intervention from their formal caregivers felt trust by experiencing a broader embedding of the intervention.

## 4. Discussion

Based on the results, we can assume that the trustful relationship between CHWs and their clients is the core mechanism of this program to obtain high satisfaction rates and to improve clients’ own perception of their psychosocial health. This trustful relationship between CHWs and citizens consists of three mental models: recognition, equality, and reciprocity. These mechanisms allow psychosocial health outcomes to evolve. Figure 3 presents a visual depiction of how the CMO configurations are linked together in an overall program theory, adapting the iceberg model.

### 4.1. Program Theory

The presence of a person who listens to concerns, acts as a soundboard for emotions, and takes time to do so, allows people to relax, as such contributing to “mental space”: the space created in one’s head when one is temporarily released from daily worries and heavy emotional luggage. This liberated mental space can be used to be fully present in the moment and draw energy from a positive experience, which can add to renewed courage to face a challenging situation. An important facilitating factor is the presence of a CHW who can focus attention on the person of the client and adapt to the client’s personal needs and boundaries. Also, feeling supported by a coordinating team enables CHWs to perceive themselves as capable of handling their clients’—often complex—luggage. The main mechanism identified is recognition. Clients feel reassured by the fact that their CHW has a broad view on and knowledge and understanding of clients’ living environment, and they feel understood and accepted as a person. This is in line with the findings of two recent reviews [13,36] on CHWs and mental health, which described the added value of CHWs’ commitment to alleviating the acute need for psychosocial support during a health crisis and beyond.

When CHWs and clients have some life experiences in common, or just share and exchange personal experiences, it may reduce feelings of loneliness, help put clients’ problems into perspective, or stimulate participants to find the strength to improve their own situation. As such, they can serve as role models for clients. This relates to existing literature describing how peers can serve as community role models and can help to empower vulnerable citizens [16]. Helpful (for connecting) is the CHW being familiar with the social vulnerabilities experienced by the clients. Having a similar socioeconomic background is not an essential condition. However, this enables CHWs to feel confident in their task. An important facilitator is the temporary character of the contact, as sharing personal background and interacting with vulnerable people facing multiple challenges can be perceived as demanding for CHWs.

Because clients are provided the opportunity to develop a real connection, from human to human, and to get to know an unprejudiced volunteer, who does not have a professional agenda, they perceive a sense of belonging and improved self-worth because they feel recognized as a member of the community and they feel acknowledged in themselves. This concept of connectedness has been recognized previously as an important outcome of CHW interventions [10,16]. Clients with a background of vulnerability feel part of a group, a bigger entity; they feel noticed and known (‘someone remembers your name’) by CHWs and by their professional caregivers. More specifically, the link with primary care practice is an important facilitator for a trustful relationship to evolve between CHWs and clients. As well as for CHWs as for clients, this link appears to build trust that the right people for the right reasons are referred to the CHWs.

**Figure 3 ijerph-19-02496-f003:**
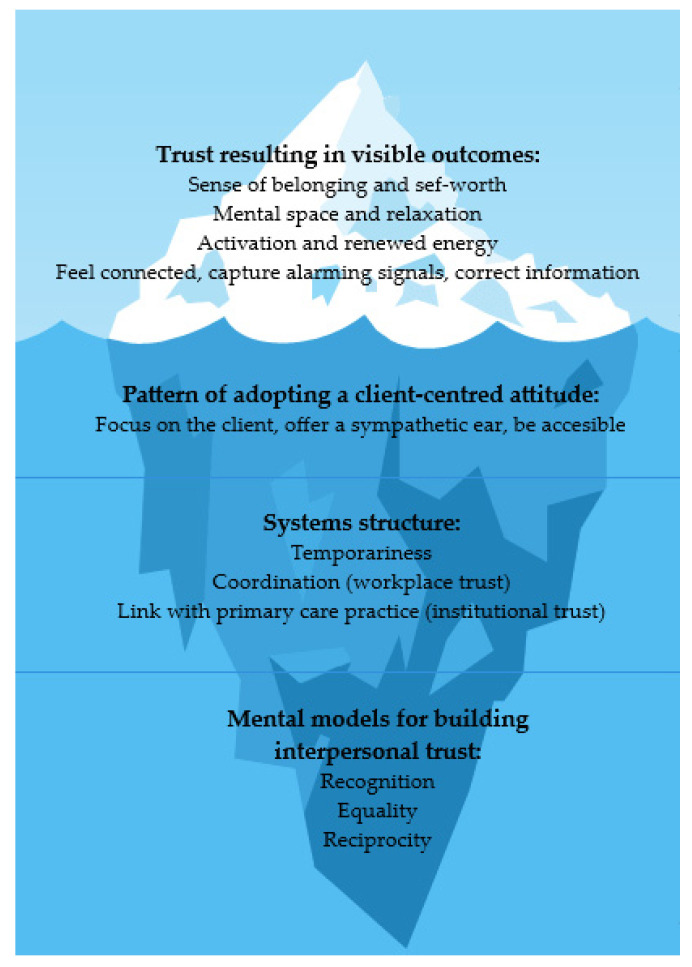
Program theory presented in the iceberg model.

### 4.2. The Role of Trust in Primary Healthcare

The role of trust in health care has traditionally been examined in relation to doctor–patient relationships. However, recent literature has sought to bring social relations of trust to the center stage in studying health systems and policies [19,23,24]. Drawing on empirical evidence from a number of contexts [18,26,27], these studies have demonstrated that trust matters to health systems. CHWs in this study reinforced this conviction. These CHWs espouse an integrated approach to care by fostering relations of mutual trust, addressing community health and other needs, promoting a continuum of care from curative to preventive care, and valuing the role of regular and effective communication with citizens and amongst health workers themselves. These values are indeed the cornerstone of a primary health care ideology that promotes equity and participation.

In vulnerable communities, where people experience structural barriers to healthcare [37], CHWs have to navigate complex health and social situations. Trust, essential in all patient–provider–health system relationships, is more fragile in such communities [24]. Patients who have unstable lives often receive poorer care; in turn, poor quality care may cause patients to lose trust in their local facility and become reluctant to seek care in the future [12]. CHWs, responsible for bringing people (back) into care, must repair that trust. This is a complex task, especially when accessibility problems and challenges to reach out to vulnerable people were even more pronounced during the COVID-19 pandemic with its accompanying physical distancing and lockdown measures [18]. The specific mental models of a CHW–client trustful relationship complement the patient–provider trustful relationship. First, recognition as a mechanism is very distinct from formal caregivers’ professional empathy since they mostly do not share a similar background of vulnerability with their patients. As such, most professional caregivers have not lived through the complexity of vulnerability themselves. Because professional caregivers have a professional task to fulfill, they mostly have their own ‘health agenda’. This hierarchical position is opposite to the contact, from human to human, between CHWs and community members, which is based on equality. Sharing your personal story, on the basis of reciprocity, is a major asset for building trust. Professional caregivers mostly keep a professional distance. By sharing stories, however, participants in this study mentioned feeling connected.

By offering a real human presence, we were able to work towards a sense of belonging for vulnerable people living on the margins of our society. On the other hand, CHWs have the potential to interconnect different communities and could possibly serve as a liaison between community and professional healthcare. In this way, the interpersonal trust between CHWs and clients can contribute to institutional trust. 

### 4.3. Strengths, Limitations, and Future Research Opportunities

Our realist and grounded theory-building approach allowed enhanced data validity and reliability. Data were collected and analyzed in practice, in real-life settings. In this study, we collected qualitative data on a CHW support program in the context of the city of Ghent during the COVID-19 pandemic. Since controlling the variables is not possible when studying complex social problems, it is important to know as much as possible about the variable in which the supposed key mechanisms function. Therefore, keen documentation of the context in which the mechanism is trigged is required, and this is preferably repeated in differing contexts and circumstances. Although the selection of the cases and the interviewees of interest has been done with respect to this principle, problems according to the generalizability of the findings could still arise. The context of the COVID-19 pandemic in this study could possibly impact the generalizability of our findings since these circumstances themselves could already influence the process of building trust between CHWs and clients. Possibly, clients were more suspicious towards volunteers in these challenging times with anxiety prevailing, or, by contrast, clients may be more eager to trust volunteers in times of social isolation. Further research in other—non-COVID—contexts is needed to examine whether this theory can be consolidated.

Another limitation of this study could be the strong involvement of the researcher D.V.B. in the program, which could risk assumptions being made beforehand. To overcome this, the data were triangulated with the two other researchers, P.D. and S.W.

One of this study’s main strengths, namely the program theory being partly grounded in data and not solely the result of a theoretical exercise, has generated some limitations as well. Our program theory reflects the idea that context elements at the micro-level (adopting a client-centered attitude, coordination, temporariness, and link with primary care practice) indeed play a huge role as a catalyzer for key mechanisms. However, the “upstream” social determinants of health, such as social disadvantage, risk exposure, and structural inequities, also play a fundamental role. Context elements at meso- (organization, network, partnerships, local politics, …) and macro-level (policy, law, regulation, ...) may trigger or impede important context elements at the micro-level. Therefore, more attention is needed for the cascade of context factors at the structural (political and societal) level allowing (or impeding) these mechanisms.

In this study, the lack of affiliation with an institution and the status of a CHW as a nonprofessional healthcare provider increases the building of trust with community members. However, it is also important to explore CHWs’ role in relation to other health workforce and to integrate CHW interventions into the general health and community system [14]. Through their specific backgrounds and interactions with community members, CHWs develop expert knowledge, a nuanced health ‘intelligence’ that risks being lost when not integrated into the primary care team. Unfortunately, strategies that elaborate on systems of CHWs allowing community members’ needs and visions to be actively involved in the organization and operation of PCP remain scarce. This points to the need for co-creation of knowledge and for active involvement of communities, which has not yet been constructed. This is an ambition for the future.

### 4.4. Study Implications and Recommendations for Policymakers and Practitioners

As confirmed by other lines of evidence, CHWs can be effectively engaged to provide psychosocial support at the community level [12,14]. Engaging them during times of crisis can also be cost-saving as they have already been demonstrated to be a less expensive alternative compared to other health professionals, and little extra effort is needed to recruit or engage them as many of them are already employed [14]. However, they need adequate training and supervision, and their safety and security must be protected, especially during this COVID-19 pandemic.

CHWs need to be equipped with adequate training before engaging in providing psychosocial support. They need to be trained in assessment, communication skills, problem-solving, professional responsibilities and boundaries, as well as stress and emotion management strategies. Previous research has demonstrated that short duration training for up to 2 weeks produced good results [13]. However, building trust with vulnerable community members remains a complex skill requiring multiple competencies. Therefore, training and surveillance of CHWs’ competencies should be an ongoing process, requiring individual and in-group guidance by a supporting, coordinating, and supervising team that solves problems and improves skills. Moreover, management of CHWs requires sustainable support by and integration into local and national health systems, plans, and policies. In order to keep CHWs motivated to continue their training and improve their skills, a financial reward should be considered, as this could also facilitate integration in the primary care team.

Professional caregivers should be aware of the importance of trust as a key mechanism for CHWs to reach out to community members. As for other healthcare workforce, this specific trustful relationship should be valued as complementary to gain trust at interpersonal and at institutional levels. Therefore, professional caregivers should be cautious not to ‘use’ CHWs as a strategy to reach their health goals and should respect this extra dimension, so CHWs do not feel ‘caught in the middle’ between reaching individual patients’ health goals and reaching out to vulnerable citizens.

## 5. Conclusions

This realist and grounded theory-building approach to a CHW program has shown that CHWs may be the connection between some vulnerable citizens and an ever-changing health care system during the COVID-19 pandemic. More specifically, CHWs present as a crucial public health outreach strategy for PCP by complementing and enhancing trust-building by primary care professionals. In this process of building trustful relationships between CHWs and clients, different mechanisms and contextual factors play a role in the trustful relationships between primary care professionals and patients. In addition, this identification of key mechanisms and contextual factors for building trustful relationships between CHWs and vulnerable citizens allows program developers to consider essential working ingredients and contextual boundaries in setting-up successful CHW interventions in HIC primary care settings, during COVID-times and beyond. Further research is needed to determine whether the same mechanisms and contextual factors apply in other situations. More specifically, it should be assessed whether these findings also apply to a non-covid context, to the involvement of CHWs in other facets of primary healthcare (e.g., prevention campaigns, etc.), and to a low- and middle-income country (LMIC) setting. Furthermore, implementation research should elaborate on the integration of CHWs in PCP, to support CHWs in developing the mental models leading to build trust with vulnerable citizens, and to establish the required conditions.

## Figures and Tables

**Figure 1 ijerph-19-02496-f001:**
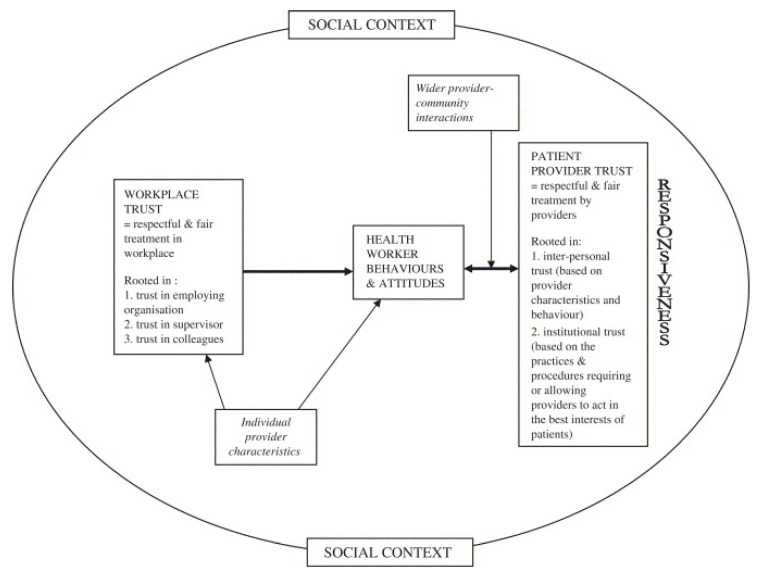
Conceptual framework on trust from Gilson et al. [27].

**Figure 2 ijerph-19-02496-f002:**
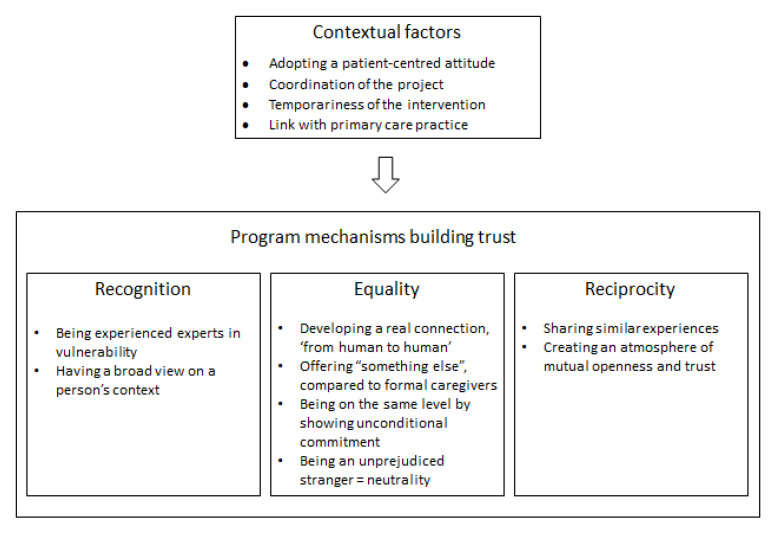
Contextual factors and program mechanisms producing outcomes.

**Table 1 ijerph-19-02496-t001:** Sociodemographic characteristics for participants of focus group discussions and in-depth interviews. FGDs (focus group discussions).

		FGDs	In-Depth Interviews	
		CHWs	CHWs	Clients
		*N* = 8 (%)	*N* = 13 (%)	*N* = 11 (%)
Sex	Male	3 (37.5%)	3 (23.1%)	4 (36.4%)
	Female	5 (62.5%)	10 (76.9%)	7 (63.6)
Age (yrs)	<25	0	1 (7.7%)	0
	25–39	3 (37.5%)	7 (53.8%)	1 (9.1%)
	40–64	5 (62.5%)	5 (38.5%)	4 (36.4%)
	≥65	0	0	6 (54.5%)
Work or activity	Student	0	2 (15.4%)	1 (9.1%)
	Worker/Servant/Self-employed	4 (50%)	6 (46.1%)	0
	Job-seeking	1 (12.5%)	2 (15.4%)	1 (9.1%)
	Retired	2 (25%)	2 (15.4%)	6 (54.5%)
	Disability	1 (12.5%)	1 (7.7%)	3 (27.3%)
Migration background	No	8 (100%)	9 (69.2%)	7 (63.6)
	Yes,			
	Living <1 year in Belgium	0	0	1 (9.1%)
	Living 1–5 years in Belgium	0	0	0
	Living 6–10 years in Belgium	0	1 (7.7%)	1 (9.1%)
	Living >10 years in Belgium	0	3 (23.1%)	2 (18.2%)

## Data Availability

The data presented in this study are available on request from the corresponding author. The data are not publicly available due to ethical considerations and privacy restrictions.

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
