# Peer review of "Understanding Trustful Relationships between Community Health Workers and Vulnerable Citizens during the COVID-19 Pandemic: A Realist Evaluation"

_ijerph, 2022, doi:10.3390/ijerph19052496_

Round 1

Reviewer 1 Report

Title: missing during the COVID-19 pandemic

Abstract:

Background: it is necessary to better identify the rationale to underline.

Methods: Why do you rewrite the purpose that is different from the one indicated in the background? (line 13-14).

Results: “how trustful relationships are built” is unclear.

Conclusions: generic and do not give indications. Also we talk about professional caregivers, not mentioned in the background.

Keywords: grounded theory; professional caregiver?

Background:

Too generic. It is suggested to better focus on the rationale to be emphasized.

We start by talking about the epidemic of infectious diseases and their effects (line 28), when this topic is not indicated in the title, in the abstract, or in the keywords.

The Covid-19 pandemic (line 32) and psychosocial effects (line 34) are discussed below, when not specified in the title and only mentioned in the abstract method, and Ebola virus (line 50)?.

It would be appropriate to initially focus attention on trustful relationships (definition and literature data) between CHWs and vulnerable citizens.

It is suggested to summarize the information from the studies conducted on the role of CHWs in public health reported in lines 53-80 of the manuscript. Such data in detail could be more useful and appropriate in discussions.

Here the aim is different, in fact it introduces "with community members in high-income country (HIC) primary care settings ? (line115).

The objective could be formulated more accurately and with terms more suited to qualitative research.

Materials and methods:

The description of the pilot project conducted in Ghend presented at the beginning of this section is a bit confusing, in which we should in fact speak exclusively of the study presented in the article. From lines 123-164 it is not clear whether the description of the sample recruited and the intervention proposed in this study or in the original pilot project is presented. This ambiguity generates confusion throughout the study. It is also not clear whether the interviews and focus groups were guided by key questions to guide the conversation towards the areas of the phenomenon of interest.

Section 2.1 The study setting and characteristics of the intervention:

Only the setting should be discussed in this section. For a realistic qualitative study of this type it is not appropriate to speak of intervention. In addition, some characteristics of (incomplete) recruitment are described that should be included in the section. 2.4 Participant recruitment, data collection and data analysis.

Section 2.4 Recruitment of participants, data collection and data analysis:

The modalities and characteristics of the recruitment, the criteria for inclusion and those of a possible exclusion should be included. It is also not clear how the focus groups and interviews were structured (there were moderators, observers,… ??). A bibliographic reference on the saturation of theoretical data is missing (line 216). Also, the number of people recruited is entered in this section, which should instead be described in the results section.

Results

The socio-demographic data of the recruited sample is missing (tables and descriptions).

Section 3.1 Outcomes of the intervention: in a study of this type it is not appropriate to talk about intervention. Interviews and focus groups are in fact methods of data collection and not actual interventions.

A small paragraph on the limitations of the study would be appropriate; only the scarce generalizability? (line 527).

Incomplete discussions: The programm theory and sociodemographic data of the sample have not been commented through the literature.

Conclusions:

The objective is repeated by adding “during times of crisis” not well specified by the title.

Generic,  and how relationships are created is not clear

Reviewer 2 Report

Thank you for the opportunity to review your paper. The paper entitled “Understanding Trustful Relationships between Community Health Workers and Vulnerable Citizens: A Realist Evaluation” uses a grounded theory approach to deconstruct the trustful relationships between community health workers (CHWs) and the individuals they serve. This exploration is meaningful and could be instrumental in informing CHW programs and similar population health interventions. My recommendations for the paper are listed below-

Introduction

·       The introduction needs structural improvement. For instance, smoother transition is needed between the first and second paragraph of the introduction. Also, the presentation of the extant literature on the effectiveness of CHWs seems disjointed. I recommend the author(s) synthesize the evidence referenced and present them in a more organized fashion.

·       Line 81-86: The authors state that there is limited evidence on the attributes of the “trustful relationships” between CHWs and their clients. What related literature exists to give the readers more context and a robust understanding of the nature of the CHWs relationship-building process? What evidence exists on barriers and facilitators to trust building within the context of the CHWs-client relationship?

Materials and Methods

·       Line 207-209: Can the author(s) provide more information on how diversity was used as a selection criterion?

Results

·       The findings are intriguing and well-presented.

Discussion

·       Line 444-474: How do the findings discussed in the subsection compare to extant literature?

·       What are the limitations of the study?

Reviewer 3 Report

Dear auhtors,

I read this paper with great interest. In the covid-19 pandemic situation, social work will be even more important.
I am not an expert in this research feild. Therefore, I may not fully understand the method. The comments below are what I felt frankly.

Figure and table
There are many figures created by others, and few figures and tables created by the author. I don't think it's that important to quote someone else's figure. Rather, the authors should reconstruct the logic of the citations to create the own conceptual diagram suitable for this study.

Method
>We developed a program theory using a realist research design. Data were collected through focus groups and in-depth interviews with CHWs and their clients. Using a grounded theory approach, we aimed to unravel mechanisms and contextual factors that determine the trust in CHWs offering psychosocial support 14 to vulnerable people during the COVID-19 pandemic.

”we aimed to unravel mechanisms and contextual factors that determine the trust in CHWs offering psychosocial support to vulnerable people during the COVID-19 pandemic”
I think this should rather be written for the method.

It is difficult to verify the validity of the results without knowing what kind of interview was conducted. Is it possible to show the content of the interview guide?

Line 221
The analysis method is abstract and difficult to understand. Is it possible to make the expression a little easier to understand? 
Also, using figure may help understanding.

Result and Discussion
It was a very interesting result. However, this study aims to clarify process of creating trust between vulnerable people and CHW in the situation of covid-19. Overall, the results feel like a general discussion of the creating process of relationship between the client and CHW. I think it would be better to develop a discussion focusing on covid-19. The same is true for the conclusion. The authors mention "during the COVID-19 pandemic" for the aim of study, so I think it needs a suitable conclusion.
